# Association of Daily Sitting Time and Leisure-Time Physical Activity with Sarcopenia Among Chinese Older Adults

**DOI:** 10.3390/healthcare13030251

**Published:** 2025-01-27

**Authors:** Yujie Liu, Zhengyan Tang, Xiao Hou, Yaqing Yuan, Yunli Hsu, Jinxia Lin, Jingmin Liu

**Affiliations:** 1Division of Sports Science and Physical Education, Tsinghua University, Beijing 100084, China; liuyujie24@mails.tsinghua.edu.cn (Y.L.); crystal_267@163.com (Y.Y.); bieber84917@gmail.com (Y.H.); linjx23@mails.tsinghua.edu.cn (J.L.); 2Department of Physical Education, Southeast University, Nanjing 210096, China; tzy0565@sina.com; 3School of Sport Science, Beijing Sport University, Beijing 100084, China; houxiao0327@bsu.edu.cn; 4College of Sports and Health, Shandong Sport University, Jinan 250102, China

**Keywords:** sarcopenia, daily sitting time, physical activity, old adults

## Abstract

Objectives: This study aimed to explore the independent and joint associations of daily sitting time and leisure-time physical activity (LTPA) with sarcopenia among older adults. Methods: The participants were 847 community-dwelling adults aged 60 or older from Beijing and Shanghai, China. Sarcopenia was diagnosed based on the criteria established by the Asian Working Group for Sarcopenia (2019). Daily sitting time and LTPA were self-reported using the Physical Activity Scale for the Elderly (PASE). Logistics regression models were used to explore the associations between daily sitting time, LTPA, and sarcopenia. To examine joint associations, participants were classified based on daily sitting time and LTPA levels. Final models were adjusted for sociodemographic variables, lifestyle factors, and chronic conditions. Results: Prolonged sitting time and insufficient LTPA were independently associated with higher odds of sarcopenia. Among insufficiently active participants, sitting for 1–2 h, 2–4 h, and more than 4 h per day was associated with 5.52-fold (95% CI: 1.13–26.83), 6.69-fold (95% CI: 1.33–33.59), and 12.82-fold (95% CI: 2.75–59.85) increased odds of sarcopenia, respectively, compared to sitting for less than 1 h. For those meeting the physical activity guideline (≥150 min of LTPA per week), only sitting for more than 4 h per day was significantly associated with higher odds of sarcopenia (OR: 7.25, 95% CI: 1.99–26.36). Conclusions: Prolonged sedentary behavior was associated with increased odds of sarcopenia. The higher odds of sarcopenia associated with more than 4 h daily sitting may not be offset by achieving the recommended levels of physical activity.

## 1. Introduction

The relationships between daily sitting time, leisure-time physical activity (LTPA), and sarcopenia among older adults have emerged as critical areas of research, especially with the increasing prevalence of sedentary lifestyles in aging populations. The World Health Organization recommends at least 150 min of moderate-intensity physical activity per week for older adults while emphasizing the need to reduce sedentary time [1]. Regular LTPA is essential for reducing the risk of chronic diseases, improving mental health, and enhancing overall quality of life [2,3]. Conversely, excessive sedentary behavior has been linked to increased risks of cardiovascular diseases and other health complications [4]. Despite these well-established benefits, many older adults fail to meet physical activity guidelines due to physical limitations, lack of motivation, and environmental barriers [5]. Furthermore, due to the limited evidence, no specific quantitative recommendations on sedentary behavior have been established [6].

Sarcopenia, characterized by the progressive loss of skeletal muscle mass and function, is a prevalent condition among older adults globally [7], which poses significant health risks for older adults, including increased vulnerability to falls, disability, and mortality [8,9]. According to the China Health and Retirement Longitudinal Study (CHARLS), sarcopenia affects approximately 12.8% of older Chinese adults, with a higher prevalence observed in rural areas [10]. Recent evidence has highlighted the roles of lifestyle behaviors, such as insufficient physical activity, prolonged sitting time, and poor dietary patterns, as risk factors for sarcopenia [11]. While moderate-to-high intensity LTPA has been inversely associated with sarcopenia prevalence and positively linked to muscle strength and bone health, sedentary behavior, particularly prolonged television viewing, has shown detrimental effects on body composition [12]. Despite these findings highlighting the importance of promoting physical activity in older adults to prevent sarcopenia and maintain healthy body composition, limited research has examined the combined effects of prolonged sedentary behavior and LTPA on sarcopenia, particularly among older adults in China. While the relationship between daily sitting time and physical activity levels has been investigated in the context of various health outcomes, such as weight gain [13], cognitive function [14], and diabetes risk [15], its specific association with sarcopenia remains underexplored.

With the rapid urbanization and aging of the Chinese population, sedentary lifestyles have become increasingly prevalent, highlighting the urgent need to understand how sedentary behavior and LTPA interact to influence sarcopenia risk. Unlike occupational or transportation-related physical activities, LTPA is voluntary and adaptable, making it a practical target for public health interventions. Therefore, this present study aims to address these gaps by investigating the independent and joint associations of daily sitting time and LTPA with sarcopenia among a sample of Chinese older adults.

## 2. Materials and Methods

### 2.1. Study Design and Participants

This study was a cross-sectional analysis conducted among community-dwelling older adults in Beijing and Shanghai, China. This study aimed to investigate the risk factors for sarcopenia and frailty in older adults. Participants were eligible if they were aged ≥60 years, had clear consciousness, possessed basic reading and writing abilities, and did not have significant speech or hearing impairments.

Participants were recruited using a simple random sampling method from community resident registries provided by local community centers. Recruitment and testing were conducted over a two-month period. A total of 1200 individuals were invited to participate, and 928 agreed (response rate: 84.4%). After excluding those with incomplete data or who did not meet the inclusion criteria, 847 participants were included in the final analysis.

Data collection involved a comprehensive process, including face-to-face interviews and physical measurements, all of which were completed in a single day. The questionnaire collected information on sociodemographic characteristics, lifestyle factors, chronic conditions, physical activity behaviors, and other relevant variables. Physical measurements included assessments of body composition, muscle strength, and physical performance, performed using standardized protocols and equipment. All data collection procedures were conducted by trained researchers to ensure accuracy and consistency. Written informed consent was obtained from all participants prior to their inclusion in this study, and ethical approval was granted by the Institutional Review Board of Tsinghua University Science and Technology Ethics Committee (Approval Number: THU01-20240130).

### 2.2. Assessment of Daily Sitting Time and LTPA

Self-reported daily sitting time and weekly LTPA were assessed through participant interviews using the Physical Activity Scale for the Elderly (PASE). The PASE was developed in 1993 by Washburn and colleagues at the University of Illinois at Urbana-Champaign. It is a standardized self-reported questionnaire designed to assess physical activity levels in older adults. The PASE consists of three main domains: leisure-time physical activities, household activities, and work-related activities, encompassing a total of 13 items [16]. It has been widely used in research across various countries and has demonstrated good reliability and validity. In the Chinese population, its reliability and validity have been tested, with a reported reliability coefficient of 0.897 and a validity coefficient of 0.442 [17].

Leisure physical activity includes activities such as sedentary behavior, walking, low-intensity physical activity, moderate-intensity physical activity, vigorous-intensity physical activity, and resistance training. Each activity is scored based on its frequency and duration using standardized categories. Frequency (“days of activity per week”) is categorized as never = 0, 1–2 days = 1.5, 3–4 days = 3.5, and 5–7 days = 6. Duration (“time of activity per day”) is categorized as less than 1 h = 0.5, 1–2 h = 1, 2–4 h = 3, and more than 4 h = 5.LTPA (minutes/week) = frequency × duration × 60(1)

Daily sitting time was categorized into four groups: “<1 h/day”, “1–2 h/day”, “2–4 h/day,” and “>4 h/day”, based on the PASE questionnaire responses. LTPA was converted into hours per week in reference to the Physical Activity Guidelines for the Chinese Population (2021) and further categorized into three groups: “<150 min/week”, “150–300 min/week”, and “>300 min/week” [18].

### 2.3. Diagnosis of Sarcopenia

The diagnostic criteria for sarcopenia in this study followed the guidelines of the Asian Working Group for Sarcopenia (AWGS 2019) [19]. Sarcopenia was defined as low muscle mass accompanied by either low muscle strength or poor physical performance [20]. The specific measurements and thresholds were as follows [19]:

(1)Muscle Mass

Muscle mass was assessed using bioelectrical impedance analysis (Model: BCA-2A, TFHT, Beijing, China). Low muscle mass was defined as an appendicular skeletal muscle mass index (ASMI) of <7.0 kg/m^2^ for men and <5.7 kg/m^2^ for women.ASMI (kg/m^2^) = appendicular skeletal muscle mass/height^2^(2)

(2)Muscle Strength

Handgrip strength was measured using a digital hand dynamometer (Model: CSTF-WL, TFHT, Beijing, China). Participants stood upright with their arms naturally hanging down. Measurements were taken from the dominant hand, as self-reported by the participants. Each participant performed two trials with a 1 min rest between trials to prevent fatigue. The maximum value from the two trials was recorded. Low muscle strength was defined as <28 kg for men and <18 kg for women.

(3)Physical Performance

Gait Speed: Walking speed was measured over a 6 m course, and a gait speed of ≤1.0 m/s was considered indicative of poor physical performance.

### 2.4. Assessment of Covariates

Covariates included sociodemographic data (age, sex, ethnicity, education level, and employment status), lifestyle factors (smoking status, alcohol consumption, and dietary habits), and chronic conditions (hypertension, hypercholesterolemia, history of diabetes, cardiovascular diseases, cancer, and depression, among others). Ethnicity was categorized as Han and others. Education level was classified into four groups: primary school and below, junior high school, senior high school, and college and above. Smoking status and alcohol consumption were categorized as never, past, and current. Dietary habits were classified into three groups: vegetarian-based, balanced vegetarian and non-vegetarian, and non-vegetarian-based.

### 2.5. Statistical Analysis

Logistics regression models were used to estimate odds ratios (ORs) and 95% confidence intervals (95% CIs) for the associations between daily sitting time, LTPA, and sarcopenia. Both daily sitting time and LTPA were included as exposures in the same model, with sarcopenia (0 = “No”, 1 = “Yes”) as the outcome variable. To account for confounding effects, the final model was adjusted for age, sex, ethnicity, education level, employment status, smoking status, alcohol consumption, dietary habits, and chronic conditions. To examine joint associations, participants were classified into groups based on daily sitting time and LTPA levels, and ORs with 95% CIs were calculated using logistic regression models adjusted for the same set of covariates. Statistical analyses were performed using SPSS (version 26, Tsinghua University, Beijing, China), with a two-sided *p*-value < 0.05 considered statistically significant.

## 3. Results

### 3.1. Descriptive Statistics

A total of 847 older adults (age: 60–89 years; 412 males) were included in this study, with participant characteristics stratified by sex presented in Table 1. Among the participants, approximately 9% were diagnosed with sarcopenia. Additionally, 24.3% reported sitting for more than 4 h per day, while 35.4% engaged in less than 150 min of LTPA per week. In contrast, 64.6% reported engaging in 150 min or more of LTPA in the past week.

### 3.2. Daily Sitting Time and Sarcopenia

After adjusting for sociodemographic variables, lifestyle factors, and chronic conditions, prolonged daily sitting time was significantly associated with increased odds of sarcopenia (Table 2). Especially, compared to individuals who sat for less than 1 h per day, those sitting for 1–2 h and 2–4 h per day were associated with 5.19-fold (95%CI: 1.89–14.25) and 4.92-fold (95%CI: 1.70–14.20) higher odds of sarcopenia, respectively. Notably, participants who reported sitting for more than 4 h per day were associated with 10.93-fold (95%CI: 4.07–29.37) increased odds of sarcopenia.

### 3.3. LTPA and Sarcopenia

Being physically active was associated with reduced odds of sarcopenia (Table 2). Older adults engaging in 150–300 min of LTPA per week were associated with a trend toward lower odds of sarcopenia (OR: 0.54, 95%CI: 0.29–1.03) compared to those with insufficient LTPA (<150 min per week). Additionally, individuals who participated in more than 300 min of LTPA per week were significantly associated with reduced odds of sarcopenia (OR: 0.33, 95%CI: 0.18–0.62). These associations remained robust after controlling for sociodemographic variables, lifestyle factors, and chronic conditions.

### 3.4. Daily Sitting Time and LTPA with Sarcopenia

In the analysis stratified by LTPA, sitting for more than 4 h per day was significantly associated with increased odds of sarcopenia regardless of whether older adults were physically insufficiently or sufficiently active (Table 3). Among those who were insufficiently active, daily sitting for 1–2 h, 2–4 h, and more than 4 h were associated with 5.52-fold (95% CI: 1.13–26.83), 6.69-fold (95% CI: 1.33–33.59), and 12.82-fold (95% CI: 2.75–59.85) increased odds of sarcopenia, respectively, compared to sitting for less than 1 h per day.

For individuals engaging in at least 150 min of LTPA per week, the negative effects of sitting time appeared to be partially mitigated. In this group, only sitting for more than 4 h per day was associated with significantly increased odds of sarcopenia (OR: 7.25, 95% CI: 1.99–26.36) compared to sitting for less than 1 h per day.

## 4. Discussion

This cross-sectional study explored the associations between daily sitting time, LTPA, and sarcopenia among older adults in urban China. The findings revealed that approximately one-fourth of participants reported sitting for more than 4 h daily, while over one-third did not meet the recommended 150 min of weekly LTPA. Prolonged sitting time was independently associated with increased odds of sarcopenia, regardless of LTPA levels. In joint analysis, combinations of prolonged sitting and low LTPA were associated with increased odds of sarcopenia. Achieving the physical activity guideline of more than 150 min of weekly LTPA did not appear to entirely mitigate the adverse effects on sarcopenia of sitting for more than 4 h per day. Nevertheless, old adults who met the guideline had substantially lower odds of sarcopenia than those who did not.

This study highlights the independent and combined associations of prolonged sedentary behavior and LTPA with sarcopenia in older adults. Unlike prior studies conducted in Western populations, this study provides evidence within the context of urban-dwelling older adults in China. Physiological mechanisms underlying sarcopenia include neuromuscular junction insufficiency, impaired capillary blood flow, reduced muscle regeneration capacity, inflammation, oxidative stress, and mitochondrial dysfunction [21]. Higher levels of physical activity, cardiorespiratory fitness, and muscular strength are associated with lower odds of sarcopenia, while sedentary time showed detrimental effects [22,23]. The significant associations observed in this study suggest that reducing sedentary time, particularly among individuals sitting for more than 4 h daily, may be critical for sarcopenia prevention. Previous research indicates that sitting for more than 7 h per day increases the odds of sarcopenia (OR: 1.98, 95% CI: 1.09–3.59) [11], with each additional hour of sedentary time linked to a 6% increase in sarcopenia odds (OR: 1.06, 95% CI: 1.04–1.10) [24]. While there is no consensus on the precise threshold for prolonged sitting time, the adverse effects of sedentary behavior on muscle health are well documented. Sedentary time is associated with reduced muscle protein synthesis, neuromuscular stimulation, and mitochondrial function, which collectively accelerate muscle atrophy and functional decline [25]. Prolonged inactivity exacerbates metabolic dysregulation, insulin resistance, and shifts in substrate utilization that favor carbohydrate oxidation over fat, impairing muscle metabolism and regeneration [26,27]. Additionally, sedentary behavior is associated with increased body fat and visceral fat, further aggravating metabolic dysfunction and accelerating muscle loss [27]. These processes collectively lead to muscle atrophy, functional decline, and a heightened risk of sarcopenia. While some studies did not observe a direct correlation between sitting time and muscle strength, they acknowledged the broader implications of an increasingly sedentary lifestyle on public health [28].

Conversely, regular LTPA was significantly associated with lower odds of sarcopenia. Unlike occupational or transportation-related activities, LTPA involves voluntary and structured activities, which are more adaptable and practical for urban older adults [29]. The findings suggest that engaging in more than 300 min of LTPA per week is associated with a 67% lower likelihood of sarcopenia compared to performing insufficient activity (<150 min per week). Notably, the protective effect of LTPA remained significant after adjusting for sociodemographic and health-related factors, underscoring the robustness of the association. The role of physical activity in maintaining muscle health and preventing sarcopenia is well established, primarily through mechanisms such as enhanced protein synthesis, improved neuromuscular function, and optimized metabolic health [30,31]. Physical activity activates key pathways, such as PGC1-α, which regulate genes involved in mitochondrial biogenesis and function, thereby improving muscle endurance and mitigating the progression of sarcopenia [32]. Additionally, exercise-induced myokine release has been shown to counteract muscle deterioration associated with sarcopenia [33]. Physical activity also stimulates muscle protein synthesis and activates satellite cells, which play a crucial role in muscle repair and growth [34]. Resistance training, as part of LTPA, has been highly effective in increasing muscle mass and strength through mechanisms such as enhanced protein synthesis and muscle fiber hypertrophy [35]. Consistent with previous research, older adults with higher levels of physical activity demonstrated significantly reduced odds of sarcopenia, reinforcing its protective role against this condition [36]. Conversely, insufficient LTPA (≤150 min per week) was associated with increased odds of sarcopenia [37]. Even moderate levels of LTPA (150–300 min per week) showed a trend toward reduced sarcopenia odds, suggesting that adherence to current physical activity guidelines offers protective benefits. A study of 5418 older Chinese adults found that those who met the recommendations for moderate-to-vigorous physical activity combined with adequate fruit and vegetable intake had a significantly lower risk of sarcopenia compared to those who did not meet any recommendations [38]. These findings further highlight the importance of integrating physical activity with other healthy lifestyle behaviors to mitigate sarcopenia risk and promote musculoskeletal health in aging populations.

While previous studies have examined the relationship between physical activity and sarcopenia, few have considered the joint effects of sedentary behavior and LTPA. For example, a recent meta-analysis reported that sedentary time independently increases the risk of sarcopenia, but its interaction with different levels of LTPA remains underexplored [39]. Our joint analysis of LTPA and sedentary time indicated that high levels of physical activity were associated with a partial attenuation of the adverse effects of prolonged sitting. However, these effects were not entirely eliminated. This finding aligns with the activity-imbalance hypothesis, which suggests that the detrimental effects of sedentary behavior cannot be fully counteracted by adherence to physical activity guidelines alone [40]. Sedentary behavior is independently positively associated with sarcopenia in older adults, even among individuals with sufficient physical activity [39]. For instance, interventions like the Frail-LESS program have demonstrated that reducing sedentary time enhances muscle function and mitigates sarcopenia risk [41]. Additionally, replacing sedentary time with moderate-to-vigorous physical activity (MVPA, ≥3 METS) has been associated with significantly lower odds of sarcopenia, highlighting the importance of incorporating light-to-moderate activities into daily routines [42]. Therefore, public health strategies should adopt a dual approach, emphasizing both increased physical activity and reduced sedentary time to optimize musculoskeletal health in older populations [1]. Based on this study, we recommend promoting at least 150–300 min of LTPA per week and reducing daily sitting time to less than 4 h. Additionally, community-based exercise programs should be developed to improve accessibility and adherence.

This study has several strengths. Firstly, it focuses on LTPA, a voluntary and adaptable form of physical activity, offering a more practical and targeted approach to sarcopenia prevention compared to general physical activity. Secondly, it examines both the independent and joint associations of sedentary behavior and LTPA with sarcopenia, providing a comprehensive understanding of their combined effects on muscle health. Thirdly, the use of first-hand data collection ensures higher reliability and validity compared to studies relying on public datasets. Additionally, the findings highlight the dual importance of increasing LTPA and reducing sedentary time, offering actionable insights for targeted public health interventions.

However, one important limitation of this study is the potential for reverse causality inherent in its cross-sectional design. While the findings suggest associations between prolonged sedentary behavior, leisure-time physical activity (LTPA), and sarcopenia, the temporal relationship between these factors cannot be established. For instance, individuals with sarcopenia may already exhibit reduced physical activity levels and increased sedentary behavior due to muscle weakness or functional limitations, rather than prolonged sedentary behavior or insufficient LTPA leading to sarcopenia. Future longitudinal studies are needed to clarify the causal pathways and better understand whether reducing sedentary time and increasing LTPA can effectively mitigate sarcopenia risk, or if these behaviors are primarily consequences of sarcopenia progression. While the PASE is a widely used and validated tool for assessing physical activity in older adults, its reliance on self-reported data introduces potential limitations. Recall bias may occur as participants may not accurately remember or report their activities over the past week. Additionally, subjective interpretation of activity intensity or duration can lead to variability in responses. These limitations may affect the precision of the activity assessment and introduce measurement error. Future studies could address these limitations by incorporating objective measures, such as accelerometers, to complement self-reported data. Combining subjective and objective assessments would provide a more comprehensive evaluation of physical activity levels and minimize potential bias. Finally, the urban-based sample, along with the absence of survey weights, limits the generalizability of the findings to the national elderly population, particularly those in rural or less developed areas with distinct activity patterns. Future studies should apply survey weights or recruit participants from more diverse geographic and socioeconomic backgrounds to enhance representativeness.

## 5. Conclusions

In conclusion, this study highlights the independent and joint associations of sedentary behavior and leisure-time physical activity (LTPA) with sarcopenia in older adults. Regular LTPA is associated with lower odds of sarcopenia, whereas prolonged sedentary behavior is linked to higher odds, even among physically active individuals. These findings emphasize the need for public health strategies that simultaneously reduce sedentary behavior and promote physical activity to prevent sarcopenia and support healthy aging.

## Figures and Tables

**Table 1 healthcare-13-00251-t001:** Characteristics of participants.

	No. of Participants (Weighted %)		
	All *n* = 847	Male *n* = 412	Female *n* = 435
Ethnicity			
Han	838 (98.9)	408 (99.0)	430 (98.9)
Others	9 (1.1)	4 (1.0)	5 (1.1)
Education level			
Primary school and below	73 (8.6)	32 (7.8)	41 (9.4)
Junior high school	391 (46.2)	189 (45.9)	202 (46.4)
Senior high school	219 (25.9)	114 (27.7)	105 (24.1)
College and above	164 (19.4)	77 (18.7)	87 (20.0)
Employment status			
Intellectual labor-oriented	243 (28.7)	105 (25.5)	138 (31.7)
Light physical labor-oriented	480 (56.7)	233 (56.6)	247 (56.8)
Heavy physical labor-oriented	101 (11.9)	68 (16.5)	33 (7.6)
Others	23 (2.7)	6 (1.5)	17 (3.9)
Smoking status			
Never	543 (64.1)	167 (40.5)	376 (86.4)
Past	169 (20.0)	115 (27.9)	54 (12.4)
Current	135 (15.9)	130 (31.6)	5 (1.1)
Alcohol consumption			
Never	664 (78.4)	243 (59.0)	421 (96.8)
Past	38 (4.5)	34 (8.3)	4 (0.9)
Current	145 (17.1)	135 (32.8)	10 (2.3)
Dietary habits			
Vegetarian-based	160 (18.9)	66 (16.0)	94 (21.6)
Balanced vegetarian and non-vegetarian	628 (74.1)	303 (73.5)	325 (74.7)
Non-vegetarian-based	59 (7.0)	43 (10.4)	16 (3.7)
Chronic conditions			
No	120 (14.2)	67 (16.3)	53 (12.2)
Yes	727 (85.8)	345 (83.7)	382 (87.8)
Daily sitting time (h/day)			
<1	224 (26.4)	115 (27.9)	109 (25.1)
1–2	250 (29.5)	119 (28.9)	131 (30.1)
2–4	167 (19.7)	82 (19.9)	85 (19.5)
>4	206 (24.3)	96 (23.3)	110 (25.3)
LTPA (min/week)			
<150	300 (35.4)	135 (32.8)	165 (37.9)
150–300	211 (24.9)	112 (27.2)	99 (22.8)
>300	336 (39.7)	165 (40.0)	171 (39.3)
Sarcopenia			
No	772 (91.1)	373 (90.5)	399 (91.7)
Yes	75 (8.9)	39 (9.5)	36 (8.3)
ASMI (kg/m^2^)			
Male: <7.0/Female: <5.7	77 (9.1)	40 (9.7)	37 (8.5)
Male: ≥7.0/Female: ≥5.7	770 (90.9)	372 (90.3)	398 (91.5)
Handgrip strength (kg)			
Male: <28/Female: <18	386 (45.6)	182 (44.1)	204 (46.9)
Male: ≥28/Female: ≥18	461 (54.4)	230 (55.8)	231 (53.1)
Gait Speed (m/s)			
≤1.0	737 (87.0)	370 (89.8)	367 (84.4)
>1.0	110 (13.0)	42 (10.2)	68 (15.6)

Abbreviations: LTPA = leisure-time physical activity; ASMI = appendicular skeletal muscle mass index.

**Table 2 healthcare-13-00251-t002:** Association of daily sitting time and LTPA with sarcopenia among older adults.

	Model 1 ^a^		Model 2 ^a,b^		Model 3 ^a,b,c^	
	OR (95% CI)	*p*	OR (95% CI)	*p*	OR (95% CI)	*p*
Daily sitting time (h/day)						
<1	1 (reference)		1 (reference)		1 (reference)	
1–2	5.03 (1.85–13.64)	0.002	5.24 (1.91–14.39)	0.001	5.19 (1.89–14.25)	0.001
2–4	4.84 (1.69–13.86)	0.003	5.03 (1.74–14.51)	0.003	4.92 (1.70–14.20)	0.003
>4	10.37 (3.93–27.36)	0.000	11.28 (4.20–30.25)	0.000	10.93 (4.07–29.37)	0.000
LTPA (min/week)						
<150	1 (reference)		1 (reference)		1 (reference)	
150–300	0.56 (0.30–1.04)	0.066	0.54 (0.28–1.01)	0.054	0.54 (0.29–1.03)	0.059
>300	0.35 (0.19–0.63)	0.001	0.33 (0.18–0.62)	0.001	0.33 (0.18–0.62)	0.001

^a^ Model adjusted for age, sex, ethnicity, employment status, and educational level. ^b^ Additionally adjusted for smoking status, alcohol consumption (never, former, and current), and dietary habits. ^c^ Additionally adjusted for chronic conditions. Abbreviations: 95% CI = 95% confidence interval; LTPA = leisure-time physical activity.

**Table 3 healthcare-13-00251-t003:** Association of daily sitting time with sarcopenia stratified by LTPA.

Daily Sitting Time (h/day)	Model 1 ^a^		Model 2 ^a,b^		Model 3 ^a,b,c^	
	OR (95% CI)	*p*	OR (95% CI)	*p*	OR (95% CI)	*p*
LTPA < 150 min/week						
<1	1 (reference)		1 (reference)		1 (reference)	
1–2	7.27 (1.55–34.12)	0.012	5.89 (1.21–28.72)	0.028	5.52 (1.13–26.83)	0.034
2–4	8.15 (1.65–40.26)	0.010	7.33 (1.47–36.54)	0.015	6.69 (1.33–33.59)	0.021
>4	16.82 (3.72–76.03)	0.000	14.57 (3.15–67.48)	0.001	12.82 (2.75–59.85)	0.001
LTPA ≥ 150 min/week						
<1	1 (reference)		1 (reference)		1 (reference)	
1–2	3.20 (0.87–11.75)	0.080	3.26 (0.88–12.08)	0.077	3.26 (0.88–12.09)	0.077
2–4	2.73 (0.66–11.25)	0.166	2.84 (0.68–11.87)	0.153	2.85 (0.68–11.90)	0.152
>4	6.10 (1.72–21.68)	0.005	7.24 (1.99–26.31)	0.003	7.25 (1.99–26.36)	0.003

^a^ Model adjusted for age, sex, ethnicity, employment status, and educational level. ^b^ Additionally adjusted for smoking status, alcohol consumption (never, former, and current), and dietary habits. ^c^ Additionally adjusted for chronic conditions. Abbreviations: 95% CI = 95% confidence interval; LTPA = leisure-time physical activity.

## Data Availability

The raw data supporting the conclusions of this article will be made available by the authors on request.

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
