# Peer review of "Association of Daily Sitting Time and Leisure-Time Physical Activity with Sarcopenia Among Chinese Older Adults"

_healthcare, 2025, doi:10.3390/healthcare13030251_

Round 1
Reviewer 1 Report
Comments and Suggestions for Authors
This cross-sectional study examined the associations between daily sitting time, leisure-time physical activity (LTPA), and sarcopenia. While the manuscript is generally well-written, I have a few minor suggestions for improvement:
-
Throughout the manuscript, please avoid using causal language due to the cross-sectional design of the study.
-
Methods: Section 2.1 Study Design and Participants
Please provide more detailed information on the following aspects of study participant collection: the survey aim, sample selection method, data collection procedures, study period, and response rate. -
Please include a more detailed explanation of the PASE.
-
Please consider replacing "high-intensity" with "vigorous intensity" for consistency with standard terminology.
-
Please consider including additional covariates, such as marital status, income, and employment status, in the analysis.
-
If survey weights were applied in the analysis, please ensure this is explicitly stated in the methods section.
-
Line 188-190: Please consider omitting this sentence, as causal effects cannot be inferred from cross-sectional analyses.
-
Line 214: Please revise this sentence to avoid implying a causal effect of LTPA on sarcopenia, particularly given the possibility of reverse causality.
-
Line 236-238: Similarly, please revise this section to emphasize associations rather than causal relationships.
-
Line 245: Please define the abbreviation "MVPA" (moderate-to-vigorous physical activity) the first time it appears in the manuscript.
-
Please include a thorough discussion of the potential for reverse causality as a limitation of the study.
-
Line 268-269: The claim of a causal effect in this sentence is inappropriate. Please revise accordingly.
-
Provide the reference number for Institutional Review Board (IRB) approval to enhance transparency.
Reviewer 2 Report
Comments and Suggestions for Authors
The comments are replaced by the attached PDF file.

Reviewer 3 Report
Comments and Suggestions for Authors
This is a cross-sectional study aiming to investigate the association between sitting and LTPA and sacropenia. My majors concerns are as follows:
1. In “Methods”, please clarify your sampling strategy. Did you recruit residents from several predefined communities? How do you reach out to the participants? Do you know the total number of targeted participants, or the completion rate (number of those who completed the survey/total number of targeted participants)? If you did not apply sampling, this should be mentioned in the “limitations”. The sentence “A total of 847 participants were included in the study (age: 60-89 years; 412 males and 435 females)” should be moved to the “Results” part.
2. How was the questionnaire distributed—through paper forms, an online link, or face-to-face interview?
3. How did you “ensure data reliability and validity”?
4. If your aim was to analyze the association, you may need to clarify your exposure, outcome, and consider adjusting for the confounding effects. It looks like the exposure was sitting time and LTFA, the outcome was sarcopenia, and the confounders were “age, sex, ethnicity, education level, smoking status, alcohol consumption, dietary habits, and chronic conditions”.
4. In Table 1, why did you divide the participants by sex? I would recommend to divided them by your exposure (eg. sitting time) and calculate the standardized mean difference between groups, so that you will know whether each confounder was balanced across groups.
5. In Table 2, did you adjust for LTPA when analyzing the effects of sitting time, and vice versa? I recommend including both sitting time and LTPA in the same model, along with the confounders.
and M
Round 2
Reviewer 2 Report
Comments and Suggestions for Authors
Dear Authors,
Upon evaluating the revised manuscript, I have confirmed that it has been significantly and thoughtfully improved overall.
I appreciate your efforts in enhancing the clarity and quality of the manuscript, and I look forward to seeing the contributions of your future research in this field.
Reviewer 3 Report
Comments and Suggestions for Authors
The authors have successfully addressed all my concerns. I appreciate their efforts.